# Improving the Luminescence and Stability of Carbon-Centered Radicals by Kinetic Isotope Effect

**DOI:** 10.3390/molecules28124805

**Published:** 2023-06-16

**Authors:** Zhichao Ma, Lintao Zhang, Zhiyuan Cui, Xin Ai

**Affiliations:** 1School of Materials Science and Engineering, Collaborative Innovation Center of Information Technology, Collaborative Innovation Center of Marine Science and Technology, Hainan University, No 58, Renmin Avenue, Haikou 570228, China; 20080500210020@hainanu.edu.cn (Z.M.); zhanglintao@hainanu.edu.cn (L.Z.); 2State Key Laboratory of Supramolecular Structure and Materials, College of Chemistry, Jilin University, No. 2699, Qianjin Avenue, Changchun 130012, China; zycui2021@163.com

**Keywords:** organic luminescent radical, biphenylmethyl, triphenylmethyl, deuteration, kinetic isotope effect

## Abstract

The kinetic isotope effect (KIE) is beneficial to improve the performance of luminescent molecules and relevant light-emitting diodes. In this work, the influences of deuteration on the photophysical property and stability of luminescent radicals are investigated for the first time. Four deuterated radicals based on biphenylmethyl, triphenylmethyl, and deuterated carbazole were synthesized and sufficiently characterized. The deuterated radicals exhibited excellent redox stability, as well as improved thermal and photostability. The appropriate deuteration of relevant C-H bonds would effectively suppress the non-radiative process, resulting in the increase in photoluminescence quantum efficiency (PLQE). This research has demonstrated that the introduction of deuterium atoms could be an effective pathway to develop high-performance luminescent radicals.

## 1. Introduction

Organic radicals with unpaired electrons exhibit great application prospects for their unique optical, electrical, and magnetic properties [1,2,3,4,5,6,7,8,9,10]. Especially, through the doublet emission process, the theoretical upper limit of the internal quantum efficiency for radical-based light-emitting devices can reach 100% [11,12]. In order to improve the properties of luminescent radicals, several molecular design strategies were proposed. Guo et al. designed a few triphenylmethyl derivatives with different electron-rich groups, the electron structure of which did not follow the Aufbau principle, which exhibited higher photostability and photoluminescence quantum efficiency (PLQE) [13]. Alim et al. proposed the introduction of specific groups to construct non-alternant hydrocarbon which was beneficial to improve PLQE [14]. Mattiello et al. improved the photostability of PyBTM derivatives with the introduction of four phenyl groups, which improved even further when extra methoxy groups were added [15]. With terminal benzene rings on carbazole, Matsuda et al. improved the photostability of TTM-1Cz radicals efficiently [16]. Moreover, many studies have explored the effects of different substituent groups on the properties of luminescent bi- or triphenylmethyl radicals [17,18,19,20,21,22].

In addition to incorporating different substituted groups, the substitution of isotopes on relevant C-H bonds could also influence the properties of luminescent molecules, especially PLQE and stability [23,24]. Although the steric and electronic configurations are barely influenced, isotopic replacement of the protium atom (H) by the deuterium atom (D) would enormously change the bond stretch bands and decrease the energy of vibration modes, namely the kinetic isotope effect (KIE) of H/D substitution. Usually, after being substituted by D, the non-radiative transitions of molecules would be suppressed and the rates of relevant C-H(D) bond breaking would be decreased, resulting in higher PLQE and better stability under heated or irradiated conditions, even in working light-emitting diodes. Thus, the deuteration of relevant chemical bonds has become an effective way to develop high-performance luminescent materials, as well as light-emitting diodes based on closed-shell molecules with fluorescence, phosphorescence, or thermally activated delayed fluorescence (TADF) [25,26,27,28,29]. However, up until now, the KIE on properties of open-shell luminescent radicals has not been reported. Hence, in this work, molecules based on different luminescent radical systems with different numbers of deuterated carbazole groups were synthesized to investigate the KIE on properties of open-shell radicals, especially PLQE and stability.

## 2. Results and Discussion

### 2.1. Synthesis and Structure Characterization

Based on two typical luminescent radical systems, biphenylmethyl radical and triphenylmethyl radical, we synthesized four luminescent radicals with one or two deuterated carbazole groups, (N-deuterocarbazolyl)bis(2,4,6-trichlorophenyl)methyl radical (**BTM-1DCz**), (N-deuterocarbazolyl)[4-(N-deuterocarbazolyl)-2,6-dichlorophenyl] (2,4,6-trichlorophenyl)methyl radical (**BTM-2DCz**), [4-(N-deuterocarbazolyl)-2,6-dichlorophenyl]bis(2,4,6-trichlorophenyl)methyl radical (**TTM-1DCz**), as well as bis [4-(N-deuterocarbazole)-2,6-dichlorophenyl](2,4,6-trichlorophenyl)methyl radical (**TTM-2DCz**) (Figure 1). Their non-deuterated molecules (**BTM-1Cz**, **BTM-2Cz**, **TTM-1Cz** and **TTM-2Cz**) and classical radical **TTM** were also synthesized for comparison. The synthetic route followed the methods reported in the literature by using commercial materials [30,31]. There are few C-H bonds either in the biphenylmethyl or triphenylmethyl skeleton, which means the majority of C-H bonds were deuterated in the four target radicals. Since radicals are mostly silent in NMR measurements for their spin characteristics, the molecular structure and composition of target radicals were mainly confirmed by characterization of high-resolution mass spectrometer (HRMS) (Appendix A), FTIR, and elemental analysis (EA). Unlike the reported non-deuterated molecules, similar absorption peaks around 2280 cm^−1^ resulting from the aromatic C-D bond stretch, were found in the FTIR spectra of these four radicals (Appendix A). The existence of unpaired electrons in radicals was confirmed by electron paramagnetic resonance (EPR) measurements (Appendix A), the ***g*** values around 2.0040 demonstrate the feature of single free electrons.

### 2.2. Photophysical Properties

The ultraviolet-visible (UV-Vis) absorption spectra of four radicals were measured in cyclohexane solvent (Figure 2). Most of them show three typical absorption bands. The strong absorption bands, peaking at 285 nm for **BTM-1DCz** and **BTM-2DCz**, and 290 nm for **TTM-1DCz** and **TTM-2DCz**, come from carbazole. The medium absorption bands, peaking at 375 and 387 nm for **TTM** series and **BTM-1DCz**, and 369 and 403 nm for **BTM-2DCz**, are the characteristic absorption from carbon-centered radicals. Additionally, the weak absorption bands of four radicals, peaking from 450 to 700 nm which are not identical, mainly come from the intramolecular charge–transfer states compared with previous research [30,31]. Compared to the non-deuterated molecules, there is no significant change for the UV-Vis absorption spectra since there was no obvious transformation for both steric and electronic configurations between deuterated and non-deuterated molecules. In addition, **BTM-2DCz**, the only one whose non-deuterated molecule has not been reported before, shows some differences in absorption spectrum. There is an obvious absorption band peaking at 470 nm different from **BTM-1DCz**, and slight absorption enhancement around 600 nm. That is because two carbazole groups in **BTM-2DCz** are located at different sites. One is similar to the site in **BTM-1DCz**, and the other is similar to the site in **TTM-1DCz**. Thus, there would be two possible types of intramolecular charge–transfer in **BTM-2DCz**, resulting in the changes of long-wavelength absorption compared to the spectra of **BTM-1DCz** and **TTM-1DCz**. With increasing carbazole groups, the absorption of carbazole around 280 nm clearly increases. At the same time, the ratios of absorption from charge–transfer state at long wavelengths to absorption from carbon-center radicals around 380 nm are also increased, which indicates more characteristics of the charge–transfer state in more carbazole-substituted radicals. Absorption spectra in different solvents with different polarity were also measured (Appendix A), and no obvious change could be found.

Photoluminescence (PL) spectra of deuterated radicals were measured in cyclohexane solvent (Figure 2). **BTM-1DCz** and **BTM-2DCz** exhibit weak deep-red emission peaking at 711 and 706 nm, respectively. The absolute PLQE values measured by integrating sphere are 3.0% for **BTM-1DCz** and 3.6% for **BTM-2DCz**. **TTM-1DCz** and **TTM-2DCz** exhibit bright red emission peaking at 638 and 647 nm with considerable absolute PLQE values of 78.4 and 56.7%, respectively. Similar to absorption spectra, besides slight blue-shift, there is no obvious change in PL spectra of all deuterated radicals compared to non-deuterated molecules (Figure 2). It is noteworthy that the PLQE values of **BTM-1DCz** and **TTM-1DCz** in cyclohexane are almost 1.5 times as high as the relevant non-deuterated molecules reported in the literature [30,31].

To better explore the KIE on the luminescent properties of deuterated radicals, the transient PL decays of four radicals in cyclohexane were measured, as well as the unreported relevant photophysical parameters of **BTM-2Cz**. According to the measurements, radiative and non-radiative transition rates of luminescent radicals, ***k***_r_ and ***k***_nr_, were calculated, respectively. All the relevant photophysical parameters are summarized in Table 1. From the results, the ***k***_nr_ values of **BTM-1DCz** and **BTM-2DCz** decrease significantly from 245.0 × 10^6^ and 241.8 × 10^6^ to 215.5 × 10^6^ and 214.2 × 10^6^ s^−1^ compared to the relevant non-deuterated molecules. A similar reduction could also be found for the ***k***_nr_ values of **TTM-1DCz** and **TTM-2DCz**. However, the ***k***_r_ values of these four deuterated radicals show few variations. These photophysical results indicate that the deuteration of luminescent radicals does not influence the pathway and probability of radiative transitions, but significantly influences the non-radiative transitions. Because of the KIE from deuterium atoms, the non-radiative transitions of radicals were suppressed effectively, resulting in higher PLQE. The longer fluorescence lifetimes of deuterated radicals also indicate the effective suppression of bond breaking, that is to say the excited states of deuterated radicals become more stable. No matter which radical systems are used, bi- or triphenylmethyl radicals, more deuterated carbazole groups do not yield stronger suppression, so the PLQE values of **BTM-2DCz** and **TTM-2DCz** show little increases. Meanwhile, if the ratios of ***k***_nr_ to ***k***_r_ of luminescent radicals are as larger as biphenylmethyl radical systems, the influences on PLQE from the KIE of deuteration would be weak. For luminescent radicals with more balanced ***k***_nr_ and ***k***_r_, appropriate deuteration could significantly improve PLQE values.

### 2.3. Electrochemical Properties

Cyclic voltammetry (CV) analysis was performed to study the redox characteristics of deuterated radicals. Similar to non-deuterated radicals (Appendix A), all of the results mainly show two pairs of reversible redox peaks (Figure 3), indicating the reversible redox properties of deuterated radicals. The initial reduction potentials of **BTM-1DCz** and **BTM-2DCz** are −0.89 and −0.92 V, and the initial oxidation potentials are −0.02 and −0.08 V, respectively. For **TTM-1DCz** and **TTM- 2DCz**, the initial reduction potentials are −0.93 and −0.97 V, and the initial oxidation potentials are 0.49 and 0.38 V. With more deuterated carbazole groups, the conjugation of molecules would be increased, resulting in lower redox potentials. Utilizing the measurements of redox potentials, the energy levels of singly occupied molecular orbital (SOMO) and singly unoccupied molecular orbital (SUMO) were calculated. In the order of **BTM-1DCz**, **BTM-2DCz**, **TTM-1DCz**, **TTM-2DCz**, the SOMO energy levels are −4.71, −4.65, −5.22, −5.21 V, and the SUMO energy levels are −3.84, −3.81, −3.80, −3.76 V. All the values show little differences compared to non-deuterated radicals (Appendix A).

To test the redox stability of deuterated radicals, continuous multicycle (20 cycles) CV measurements were performed. From the relevant CV curves (Appendix A), neither the potentials nor the signal intensities of the redox peaks changed clearly. These results demonstrate that the unpaired electron features and molecular structures would not change after continuous multicycle CV scan, indicating the excellent redox stability of deuterated radicals.

### 2.4. Theoretical Calculations

Density functional theory (DFT) calculations (B3LYP/6-31G(d,p)) and time-dependent density functional theory (TD-DFT) calculations (CAM-B3LYP/6-31 G(d,p)) were performed to further investigate the electronic structures and transitions of deuterated radicals. Frequency calculations were carried out to ensure that all the optimized structures were minima on the potential energy surface. Figure 4 shows the optimized ground state structures and frontier orbitals of deuterated radicals. Compared to non-deuterated radicals, the calculation results did not clearly change due to the negligible influence on electronic structures from deuteration. The frontier energy levels calculated from DFT are almost the same as the SOMO and SUMO energy levels calculated from the CV measurements (Appendix A). From the results of optimized TD-DFT calculations, the emission process of radicals, namely the transitions between the lowest doublet excited states (D_1_) and the doublet ground states (D_0_), are mainly the transitions between SUMO and β-HOMO. All the electron cloud changes of these transitions are from the donors, carbazole parts to the radical centers, demonstrating the charge–transfer feature of luminescence from these radicals. The relevant calculation results of excited states are summarized in Appendix A.

### 2.5. Thermal and Photostability

Stability is a vital property in influencing the applications of luminescent radicals in most fields. Besides the redox stability discussed above, the thermal stability was also evaluated using thermogravimetric analysis (TGA). The results show that the thermal decomposition temperatures of the deuterated radicals are raised in different degrees due to the KIE from deuteration (Appendix A). In particular, the thermal decomposition temperatures of **BTM-1DCz** and **TTM-2DCz** are about 20 °C higher than non-deuterated radicals.

Another important stability that needs to be considered for luminescent radicals is photostability. The decays of fluorescence intensity from deuterated radicals were recorded under continuous irradiation from xenon lamp and contrasted with non-deuterated radicals and **TTM** radicals. The data of these decays were fitted, and the half-life was calculated (Figure 5). The half-life values of the deuterated radicals were 5.39 × 10^3^ (**BTM-1DCz**), 1.32 × 10^4^ (**BTM-2DCz**), 8.57 × 10^3^ (**TTM-1DCz**), and 1.54 × 10^4^ s (**TTM-2DCz**), respectively, indicating excellent photostability. Even compared with non-deuterated radicals, the photostability increases around 2 to 10 times (Appendix A). These results indicate that the photostability of luminescent radicals could be notably increased by deuteration, namely the excited states of luminescent radicals would be more stable due to the KIE, which is beneficial for applications.

## 3. Materials and Methods

All chemical reagents and starting materials used in this work were purchased from ERNEGI and Xilong Science Co., Ltd. (Shanghai, China) without further purification. The synthetic routes of target radicals followed the previous literature and can be found in Appendix A.

HRMS were recorded on a Shimadzu LCMS-IT-TOF. Infrared spectra were recorded on a Bruker Tensor 27 spectrometer. Elemental analysis data were recorded on an Elementar Vario microcube spectrometer. EPR data were measured using a Bruker Instruments A320 spectrometer with 10^−5^ M concentration in cyclohexane at room temperature. The UV-Vis spectra in various solvents were measured with a Shimadzu UV-1900i UV-Vis spectrometer. The PL spectra in cyclohexane were measured with a Shimadzu RF-6000 spectrometer. The transient PL decays were recorded using an Edinburgh FLS1000 spectrometer, and the absolute PLQEs were recorded on the same instrument using the integrating sphere method. DFT and TD-DFT calculations were performed on Gaussian16 commercial software (Revision C.02) [32]. Thermal stability measurements were performed on a TA INSTRUMENTS Q600 TGA analyzer under air and a ramp rate of 10 °C·min^−1^. The CV measurements were performed using a CH Instruments CHI660E electrochemical analyzer with a glassy carbon electrode as the working electrode, a platinum wire as the counter electrode, and Ag/AgCl as the reference electrode. Redox couple ferrocenium/ferrocene was used as an internal standard. The photostability was also recorded on RF-6000 under continuous irradiation from a xenon lamp.

### 3.1. Synthesis of BTM-1DCz and BTM-2DCz

Compound **BTM-1DCz** and **BTM-2DCz** were prepared following the literature procedure (Appendix A) [30]. Purple-black solid **BTM-1DCz** and dark green solid **BTM-2DCz** were simultaneously obtained from the final step with yields of 22 and 18%, respectively.

**BTM-1DCz**. **LC-HRMS** (*m*/*z*), calculated for C_25_H_4_D_8_Cl_6_N^•^ [M]^+^: 545.9574. Found: 545.9514. **Elemental analysis**, calculated for C_25_H_4_D_8_Cl_6_N^•^ (%): C, 54.88; H, 3.68; N, 2.56. Found (%): C, 54.84; H, 3.65; N, 2.58.

**BTM-2DCz. LC-HRMS** (*m*/*z*), calculated for C_37_H_4_D_16_Cl_5_N_2_^•^ [M]^+^: 685.1044. Found: 685.1006. **Elemental analysis**, calculated for C_37_H_4_D_16_Cl_5_N_2_^•^ (%): C, 64.79; H, 5.29; N, 4.08. Found (%): C, 64.77; H, 5.26; N, 4.11.

### 3.2. Synthesis of TTM-1DCz and TTM-2DCz

Compound **TTM-1DCz** and **TTM-2DCz** were prepared following the procedure in the literature (Appendix A) [31]. Reddish-brown solid **TTM-1DCz** was obtained with 71% yield of relevant step, and dark green solid **TTM-2DCz** was obtained with 83% yield of relevant step.

**TTM-1DCz. LC-HRMS** (*m*/*z*), calculated for C_31_H_6_D_8_Cl_8_N^•^ [M]^+^: 691.9078. Found: 691.9042. **Elemental analysis**, calculated for C_31_H_6_D_8_Cl_8_N^•^ (%): C, 53.80; H, 3.20; N, 2.02. Found (%): C, 53.78; H, 3.23; N, 2.02.

**TTM-2DCz. LC-HRMS** (*m*/*z*), calculated for C_43_H_6_D_16_Cl_7_N_2_^•^ [M]^+^: 829.0517. Found: 829.0498. **Elemental analysis**, calculated for C_43_H_6_D_16_Cl_7_N_2_^•^ (%): C, 62.16; H, 4.61; N, 3.37. Found (%): C, 62.17; H, 4.59; N, 3.39.

## 4. Conclusions

In summary, four deuterated luminescent radicals were synthesized to explore the KIE on luminescent radicals. The non-radiative process could be effectively suppressed by deuteration, which is beneficial to the PLQE of luminescent radicals. In particular, the PLQE value of **TTM-1DCz** significantly increased from 53.0 to 78.4%, and its ***k***_nr_ deceased from 19 × 10^6^ to 5 × 10^6^ s^−1^. The KIE of deuteration also made the luminescent radicals more stable, including redox stability, thermal stability, and photostability. Especially compared with non-deuterated radicals, the photostability of **TTM-2DCz** increased almost 10 times. These results from this paper demonstrate that the deuteration of relevant C-H bonds would be an effective pathway to develop high-performance luminescent radicals, especially to the non-deuterated luminescent radicals with balanced ***k***_nr_ and ***k***_r_. The influence of deuteration on the properties of radical-based light-emitting diodes are also under investigation.

## Figures and Tables

**Figure 1 molecules-28-04805-f001:**
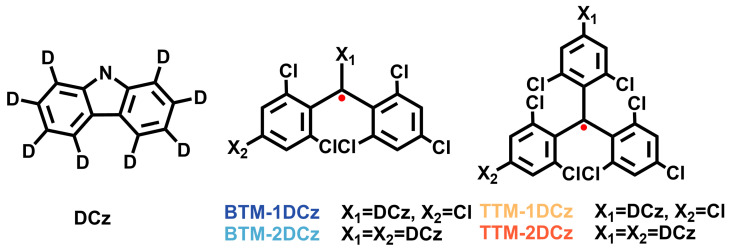
Molecular structures of **BTM-1DCz**, **BTM-2DCz**, **TTM-1DCz**, **TTM-2DCz,** and deuterated carbazole (**DCz**).

**Figure 2 molecules-28-04805-f002:**
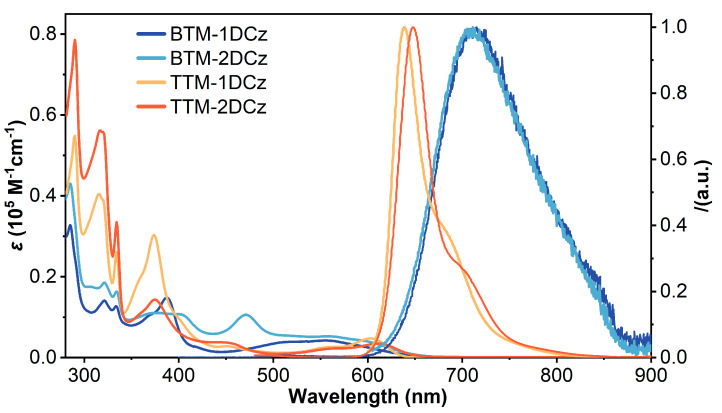
UV-Vis absorption and normalized PL spectra of four deuterated radicals in cyclohexane solvent (1 × 10^−5^ M).

**Figure 3 molecules-28-04805-f003:**
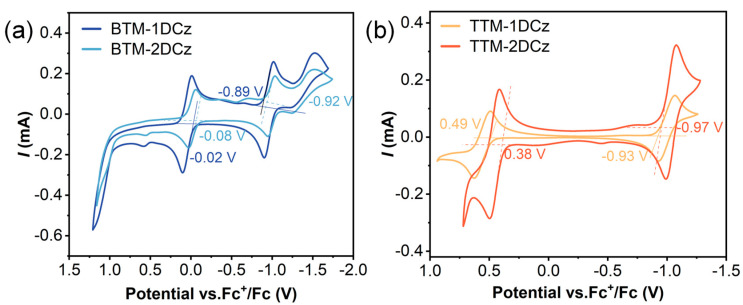
Cyclic voltammetry (CV) curves of (**a**) **BTM-1DCz** and **BTM-2DCz**; (**b**) **TTM-1DCz** and **TTM-2DCz**. Ferrocene cation/ferrocene (Fc^+^/Fc) couples were used as reference.

**Figure 4 molecules-28-04805-f004:**
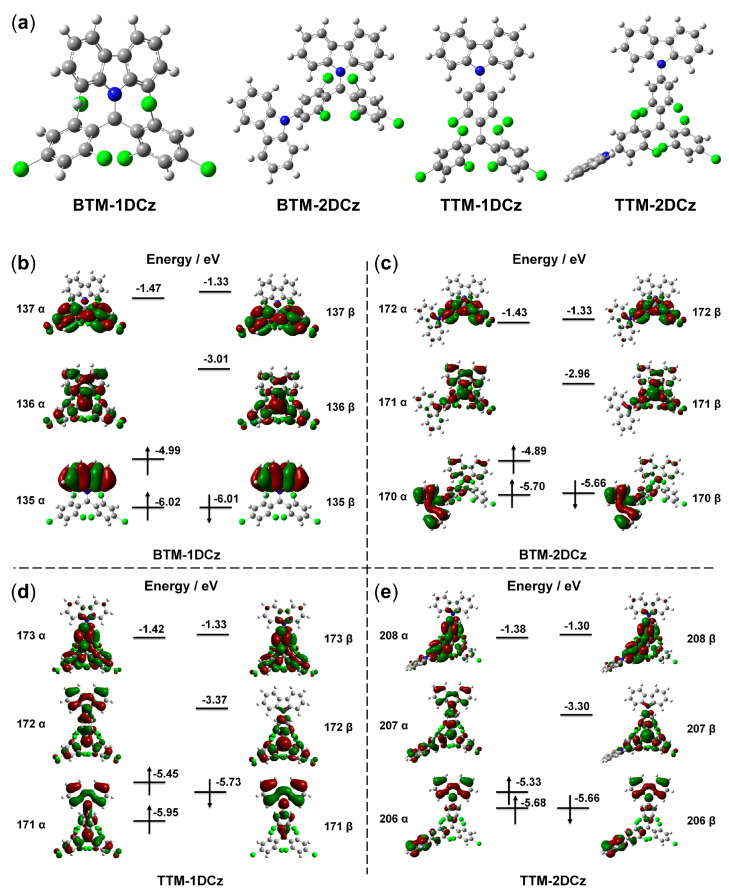
(**a**) Optimized molecular structures of deuterated radicals. The frontier orbitals of deuterated radicals (**b**) **BTM-1DCz**, (**c**) **BTM-2DCz**, (**d**) **TTM-1DCz**, and (**e**) **TTM-2DCz**.

**Figure 5 molecules-28-04805-f005:**
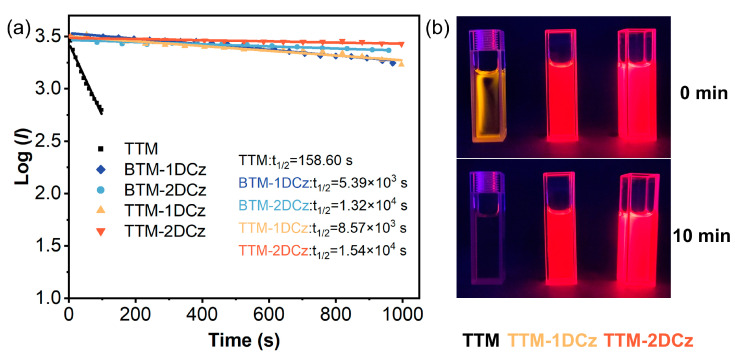
(**a**) Photostability of deuterated radicals in cyclohexane solution. (**b**) Photograph of comparison of photostability under irradiation with 365 nm handheld UV lamp.

**Table 1 molecules-28-04805-t001:** Photophysical parameters of deuterated and non-deuterated radicals.

	*λ*_Em_ (nm)	PLQE (%)	*τ* (ns)	*k*_r_ (10^6^ s^−1^)	*k*_nr_ (10^6^ s^−1^)
**BTM-1DCz**	711	3.0	4.5	6.7	215.5
**BTM-1Cz** ^a^	712	2.0 ^b^	4.0 ^b^	5.0 ^b^	245.0 ^b^
**BTM-2DCz**	706	3.6	4.5	8.0	214.2
**BTM-2Cz**	707	3.3	4.0	8.2	241.8
**TTM-1DCz**	638	78.4	41.4	19.0	5.0
**TTM-1Cz**	640	53.0 ^c^	25.3 ^c^	21.0	19.0
**TTM-2DCz**	647	56.7	33.3	17.0	13.0
**TTM-2Cz**	650	54.0 ^c^	28.0 ^c^	19.0	17.0

^a^. **BTM-1Cz**, namely **CzBTM** in ref. [30]; ^b^. Cited from ref. [30]; ^c^. Cited from ref. [31].

## Data Availability

The data presented in this study are available on request from the corresponding author.

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
