# Peer review of "Improving the Luminescence and Stability of Carbon-Centered Radicals by Kinetic Isotope Effect"

_molecules, 2023, doi:10.3390/molecules28124805_

Round 1

Reviewer 1 Report

This research encloses to provide an answer on the question: Does deuteration make influence on the photophysical properties and stability of luminescent radicals? The manuscript is a piece of pioneering work that cover very specific field – stable luminescent organic free radicals. The effect of deuteration on increasing of stability and improving photophysical properties was used for the first time. It provides completely new synthetic strategy towards stable luminescent organic free radicals with significantly improved photoluminescence. Synthetic methodology to deuterated radicals is relatively simple, but it allows to obtain compounds with useful properties. In my opinion the authors provided sufficient level of proof for the structure of deuterated radicals. I only can give advice for the future – to do the best efforts in order to grow single crystals suitable for X-ray diffraction. Scientific contents is in total agreement with common sense.

The manuscript presents very important results in the chemistry of luminescent stable organic radicals. I am strongly believe that the article deserves publication in the best journals. I have no doubts that the paper deserves publication particularly in Molecules. I did not find any significant issues in the manuscript. I think it can be accepted for publication without change. My only possible criticism is the absence of data of single crystal X-ray studies of single crystals of deuterated radicals. I hope that in future the authors will obtain single crystals and present this information for readership. Quality of figures (resolution) can be improved.

Quality of English in this manuscript is fine.

Reviewer 2 Report

Improving the Luminescence and Stability of Radicals by Kinetic Isotope Effect.

Zhichao Ma, Lintao Zhang, Zhiyuan Cui and Xin Ai

Deuteration of relevant chemical bonds is important to develop high performance luminescent materials, as well as light-emitting diodes based on closed-shell molecules with fluorescence, phosphorescence or thermally activated delayed fluorescence. Up to now, the kinetic isotope effect on properties of open-shell luminescent radicals has not been reported. This work refers to the synthesis of molecules based on different luminescent radical systems with different numbers of deuterated carbazole groups to investigate the kinetic isotope effect on properties of open-shell radicals, photoluminescence quantum efficiency and stability.

I looked at the manuscript. As far as I understand, the Authors report the synthesis of four deuterated luminescent radicals-deuterated carbazol derivatives- to explore the kinetic isotope effect on luminescent radicals. Photophysical parameters, theoretical calculations, thermal and photostability of deuterated and non-deuterated radicals are studied in detail. The organic synthesis and characterization of the deuterated carbazol derivatives are described well. This work is interesting in physical chemistry. The manuscript is well written and well documented. It is acceptable for publication as is.

Reviewer 3 Report

The manuscript entitled "Improving the Luminescence and Stability of Radicals by Kinetic Isotope Effect" as communication submitted to "molecules" is a good work and can be beneficial to readers. The word "Radiacls" in the title is too broad, and must be specified or narrowed, to reflect this piece of work. There are few minor missing in the text which need critical attention for fixing. The work done, is enough and suitable to be published in "molecules" after minor revision. Some of my specific comments are outlined hereunder,   

1. Write full name of compounds at their first mention, followed by their abbreviations as usual.  

2. Use unit (cm-1) in the text for FT-IR results instead of wavenumber. In some cases full names are not really needed owing to the well-established nature of the technique, Like NMR, FTIR, TGA etc. Authors may decide and skip some.   

 3. Lines 80--87, please assign exactly the electronic excitations occurring in molecules at the given wavelength.

4. I suggest one unit with values in a series such as -4.71 V, -4.65 V, -5.22 V, -5.21 V be replaced by -4.71, -4.65, -5.22, -5.21 V. If authors agree, then check throughout the text. 

5. "There are few C-H bonds either in biphenylmethyl or triphenylmethyl skeleton, which means majorities of C-H bonds were deuterated in the four target radicals. Since radicals are mostly silent in nuclear magnetic resonance (NMR) measurements for their spin characteristics," Here I do not pick the real concept. NMR is an informative and reliable technique, did they measure NMR (1H or 13C) for given compounds or otherwise? 

6. Check some terms such as, .....increase lesser, HRMS mass spectra, under air condition (is confusing), etc. 

7. Section 3.1, revise the wording such as, Compound XXX was prepared following the literature procedure. Revise the following sections accordingly. 

8. Add quantitative details to "Conclusion". 

 9. Figures' resolution may be enhanced to make them more clear. 

Minor polishing is required. 
